# An R1R2R3 MYB Transcription Factor, MnMYB3R1, Regulates the Polyphenol Oxidase Gene in Mulberry (*Morus notabilis*)

**DOI:** 10.3390/ijms20102602

**Published:** 2019-05-27

**Authors:** Dan Liu, Shuai Meng, Zhonghuai Xiang, Guangwei Yang, Ningjia He

**Affiliations:** Department of State Key Laboratory of Silkworm Genome Biology, Southwest University, Chongqing 400715, China; liudanpl@163.com (D.L.); xndxms@163.com (S.M.); xbxzh@swu.edu.cn (Z.X.); yanggw@swu.edu.cn (G.Y.)

**Keywords:** polyphenol oxidase, *MnMYB3R*, yeast one-hybrid

## Abstract

The aim of this study was to determine how the mulberry (*Morus notabilis*) polyphenol oxidase 1 gene (*MnPPO1*) is regulated during plant stress responses by exploring the interaction between its promoter region and regulatory transcription factors. First, we analyzed the *cis*-acting elements in the *MnPPO1* promoter. Then, we used the *MnPPO1* promoter region [(1268 bp, including an MYB3R-binding *cis*-element (MSA)] as a probe to capture proteins in DNA pull-down assays. These analyses revealed that the MYB3R1 transcription factor in *M. notabilis* (encoded by *MnMYB3R1*) binds to the *MnPPO1* promoter region. We further explored the interaction between the *MnPPO1* promoter and MYB3R1 with the dual luciferase reporter, yeast one-hybrid, and chromatin immunoprecipitation assays. These analyses verified that MnMYB3R1 binds to the MSA in the *MnPPO1* promoter region. The overexpression of *MnMYB3R1* in tobacco upregulated the expression of the tobacco *PPO* gene. This observation as well as the quantitative real-time PCR results implied that *MnMYB3R1* and *PPO* are involved in the abscisic acid-responsive stress response pathway.

## 1. Introduction

Polyphenol oxidase (PPO) is a type III copper ion protein encoded by nuclear genes, and it is widely found in animals, plants, and fungi [1]. The PPOs can be classified into the following three types depending on their substrates: Catechol oxidases, *p*-phenol oxidases, and tyrosinases [2]. In plants, most PPOs are catecholases and monophenol oxidases. Tyrosinases can hydroxylate monophenols to form o-diphenols, which are then oxidized to form the corresponding quinone. Catechol oxidases catalyze the oxidation of diphenolic compounds to form quinones [3,4]. Some catecholases are able to hydroxylate chalcones to form aurones. Thus, although catecholases cannot hydroxylate general tyrosinase substrates (e.g., tyrosine and tyramine), it has been suggested they may be able to hydroxylate other substrates [5].

Tyrosinase was first discovered in mushrooms in 1895, and since then, there has been considerable interest in PPOs among researchers worldwide. Many studies have structurally and functionally characterized PPOs. For example, previous research revealed that apple PPO is induced by wounding [6] and clarified the PPO functions possibly involved in plant defense responses. An earlier investigation proved that the leaves of tomato plants overexpressing *PPO* are resistant to *Spodoptera litura* [7]. Additionally, silencing the walnut *PPO* genes via RNA interference technology reportedly increases the chances that the resulting plants develop necrotic plaques [8]. Many other studies have confirmed the relationship between PPOs and plant defense [9,10,11,12,13,14,15,16,17]. Studies regarding the overexpression or silencing of *PPO* genes have clearly demonstrated the key role of PPOs in plant defense responses. However, the resistance mechanisms of PPOs remain unclear. The hypotheses that have been proposed for how PPOs improve plant resistance include the following three: (1) The ruthenium redox cycle leads to the production of H_2_O_2_ and other reactive oxygen species, which function as signaling molecules in the interaction between plants and pathogens; (2) hydrazine crosslinks with proteins or other phenolic substances to form a physical barrier to pathogens in the cell wall (PPO is associated with the formation of melanin-like polymers); and (3) PPO-generated quinones with toxic properties accelerate the death of pathogens or insects [16,18]. These hypotheses can explain the plant defense mechanism in terms of the downstream products of polyphenols, but the regulatory mechanisms upstream of polyphenols remain relatively uncharacterized.

The first identified plant MYB transcription factor was the protein encoded by the maize *c1* locus. This protein is involved in the regulation of anthocyanidins [19]. The largest class of MYB transcription factors in plants is R2R3MYB. These transcription factors mainly contribute to the regulation of secondary metabolism and responses to biotic and abiotic stress, and some help regulate plant growth and development. For example, AtMYB11 and AtMYB12 in *Arabidopsis thaliana* help regulate the flavonoid biosynthesis pathways [20]. The overexpression of *AtMYB75* and *AtMYB113* increases the accumulation of anthocyanins in *A. thaliana* [21]. Moreover, AtMYB30 regulates the long-chain fatty acid synthesis, thereby influencing pathogen-induced cell death [22]. Other transcription factors, such as AtMYB2, AtMYB60, and AtMYB96, control responses to drought and salt stresses through the abscisic acid (ABA) signaling pathway [23,24,25]. In *A. thaliana*, AtMYB21 and AtMYB80 are important regulators of the development of stamens and anthers, respectively [26,27]. There have been many studies on R2R3MYB, but few regarding MYB3R. To date, only a few MYB3R transcription factors have been identified. *Arabidopsis thaliana* and tobacco have five and three MYB3Rs, respectively, all of which are involved in regulating the G2/M cell phase transition [28,29,30]. The MYB3R transcription factors are highly similar to animal c-MYBs and are functional in animals and plants [31,32]. In addition to regulating the cell cycle, some MYB3Rs (e.g., TaMYB3R1 in wheat and OsMYB3R2 in rice) are involved in abiotic stress responses in plants. Previous studies confirmed that TaMYB3R1 participates in responses to drought, salt, cold, and other stresses. The overexpression of *TaMYB3R1* in *A. thaliana* affects not only plant growth and development, but also responses to osmotic stress [33,34]. The overexpression of *OsMYB3R2* in *A. thaliana* increases the tolerance to freezing, drought, and salt stresses [35], whereas its overexpression in rice alters the cell cycle turnover and enhances the tolerance to low-temperature stress [36].

In this study, we explored the roles of a transcription factor, MnMYB3R1, which regulates the *MnPPO1* gene. This transcription factor is a 3RMYB that helps regulate the cell cycle [29,37]. We observed that MnMYB3R1 also influences the expression of *PPO* genes under drought stress conditions.

## 2. Results

### 2.1. Identification of a MYB Transcription Factor Binding to the cis-element of the MnPPO1 Promoter

An analysis of *cis*-acting elements revealed a typical MYB3R-binding *cis*-element (MSA) sequence in the *MnPPO1* promoter region (Figure 1). To identify MYB proteins that bind to this MSA, we constructed a *MnPPO1* promoter probe containing the MSA sequence CAACGG for DNA pull-down analyses. When nuclear protein extracts were incubated with the biotin-labeled *MnPPO1* promoter probe, a new DNA-protein complex band on the silver-stained polyacrylamide gel was observed, this band was absent in the control group (Figure 2a). Mass spectrometry analyses were performed to identify the proteins bound to the probe (Figure 2b). We identified MnMYB3R1 as a candidate protein based on the liquid chromatography/mass spectrometry data for the biotin-labeled group (Appendix A). This protein was not detected in the non-biotinylated group (Appendix A).

### 2.2. MnMYB3R1 Binds to the MnPPO1 Promoter via the MSA

To determine whether MnMYB3R1 binds to the MSA in the *MnPPO1* promoter, yeast one-hybrid, dual fluorescence reporter, and chromatin immunoprecipitation analyses (ChIP) assays were performed. The promoter containing the MSA was inserted into the bait vector, which was transferred into yeast cells together with the *MYB3R1* expression vector. The transformed yeast cells were able to grow normally on the screening medium. However, when the MSA was mutated, the yeast cells were unable to grow on the screening medium (Figure 3a). These results were verified in the dual fluorescence reporter and ChIP analyses. The LUC/REN ratio of the experimental group was 3.2-times greater than that of the control group (Figure 3b). To verify whether the *MnPPO1* promoter binds MnMYB3R via the MSA in vivo, chromatin bound to MYB3R1 was collected with or without specific antibodies. The *MnPPO1* promoter region that could bind to the anti-MnMYB3R1 antibody was detected by a quantitative real-time polymerase chain reaction (qPCR) assay. The ChIP-qPCR results indicated that the fold enrichment of fragments 1 and 2 were five to six-times greater than that of the control, whereas the fold enrichment of fragment 3 was not significantly different from that of the control. Fragments 1 and 2 contained MSAs, but fragment 3 did not (Figure 3c).

### 2.3. Cloning of MnMYB3R1 and An Analysis of Its Expression Patterns in M. notabilis Leaves 

The *MnMYB3R1* was cloned from *M. notabilis* leaves and then sequenced. Sequence analyses indicated that *MnMYB3R1* has one 1695-bp open reading frame that encodes a polypeptide with 565 amino acids (Appendix A). Additionally, MnMYB3R1 belongs to the MYB transcription factor family and has three helix-turn-helix domains (Appendix A). In a phylogenetic tree, MnMYB3R1 clustered with OsMYB3R2 and TaMYB3R1, suggesting that these proteins may share similar functions (Appendix A). The *MnMYB3R1* transcript levels in *M. notabilis* was analyzed by qPCR. The *MnMYB3R1* transcript levels changed when the plants were infected by pathogenic bacteria or subjected to drought or salt stress. Specifically, *MnMYB3R1* expression was upregulated over time during the salt stress treatment (Figure 4a). In response to the drought treatment, the *MnMYB3R1* transcript level initially increased, peaking at 24 h and then decreased (Figure 4a). A similar trend was detected during the pathogen treatment, in which the *MnMYB3R1* transcript level peaked at 48 h after the infection and then decreased (Figure 4a). However, *MnMYB3R1* expression was upregulated within 72 h of the salt treatment (Figure 4a). The *MnMYB3R1* expression level was generally upregulated at various time-points after hormone treatments. The *MnMYB3R1* transcript levels peaked at 4 h after the salicylic acid (SA), methyl jasmonate (MeJA), and ethylene (ET) treatments. After the ABA treatment, the *MnMYB3R1* transcript level increased and peaked at 12 h (3.5-times higher than that at 0 h; Figure 4b).

### 2.4. Generation of Transgenic Tobacco Plants

A total of 16 resistant plants were obtained, and the cDNA from these plants as well as wild-type plants was used as the templates for a PCR amplification with *MnMYB3R1*-specific primers. The target sequence of the expected 1695-bp *MnMYB3R1* gene was amplified for 11 resistant plants, but not for the wild-type tobacco plants (Appendix A). These results confirmed that 11 *MnMYB3R1*-overexpressing transgenic tobacco plants had been generated. These plants were then analyzed in a qPCR assay (Figure 5a). The obtained PCR products were verified by sequencing, which indicated that *MnMYB3R1* was integrated in the tobacco genome.

### 2.5. Transcription of MnPPO1 Homologs in Tobacco

In the phylogenetic tree, MnPPO1 clustered with NtPPO2, NtPPO4, NtPPO3, and NtPPO1, and was most closely related to NtPPO2 (Appendix A). The *NtPPO1*, *NtPPO2*, *NtPPO3*, and *NtPPO4* transcript levels in the transgenic and wild-type tobacco plants were determined by qPCR. There was no significant difference in the *NtPPO1*, *NtPPO4,* and *NtPPO3* expression levels between the wild-type and the *MnMYB3R1*-overexpressing transgenic plants. However, the *NtPPO2* transcript levels differed significantly between the wild-type and transgenic plants (Figure 5b). The difference appeared to increase with increasing *MnMYB3R1* transcript levels (Figure 5b).

### 2.6. Overexpression of MnMYB3R1 Enhanced Tobacco Drought Tolerance

An analyses of *MnMYB3R1* transcript levels indicated that drought and ABA treatments induced the expression of this gene. Therefore, we examined the drought tolerance of the *MnMYB3R1*-overexpressing tobacco plants. The transgenic and wild-type plants were subjected to a two-week drought treatment by withholding water, after which they were rewatered for another week before the surviving plants were counted. The survival rates were significantly higher for the transgenic plants than for the wild-type plants (Figure 6a,b). Our results implied that *MnMYB3R1* expression was upregulated by the application of exogenous ABA. And the expression level of *MnPPO1* was also strongly induced by the ABA treatment (Appendix A). Thus, we analyzed the expression of ABA-dependent genes in transgenic and wild-type plants under drought stress conditions. With the exception of *ABI4*, the transcript levels of ABA pathway-related genes (*RD29A*, *RD29B*, *RD22*, *ABI3* and *ABI5*) were significantly higher in the transgenic plants than in the wild-type plants (Figure 6c). 

### 2.7. Determination of Polyphenol Oxidase Activity in Wild-Type and Transgenic Plants 

Under normal conditions, PPO activity was significantly higher in the transgenic tobacco plants than in the wild-type plants (Figure 6d). Additionally, the *NtPPO2* expression level was higher in transgenic tobacco plants. In contrast, there were no significant differences between the wild-type and transgenic plants regarding *NtPPO1*, *NtPPO3*, and *NtPPO4* expression levels. Therefore, the greater PPO activity in the transgenic tobacco plants compared with the wild-type plants may be related to differences in *NtPPO2* expression. Under drought conditions, PPO activity significantly increased in both transgenic and wild-type tobacco plants, but it was still higher in the transgenic tobacco plants (Figure 6d). These results suggested that PPO may influence plant drought resistance. Moreover, the *NtPPO2* expression level peaked at 24 h after the drought treatment (Appendix A).

## 3. Discussion

### 3.1. MnMYB3R1 Positively Regulates MnPPO1 Expression in M. notabilis

The MYB transcription factors are defined based on the presence of one or more MYB domain, designated as “R” [38,39,40]. Each “R” comprises about 50 amino acid residues, with a tryptophan residue distributed approximately every 18 amino acids. These conserved amino acid residues cause the MYB domain to form a helix-turn-helix structure [41]. The *MYB3R1* gene encodes an R1R2R3 MYB transcription factor that regulates the cell cycle and responses to abiotic stresses. Specifically, MYB3R regulates the cell cycle by the binding to the MSA in the promoters of genes related to the G2/M phase transition [42]. Thus, genes that have MSAs in their promoters represent candidate targets for MnMYB3R1 transcription factors.

In this study, bioinformatics and sequence analyses revealed an MSA in the *MnPPO1* promoter, and its functionality was confirmed in DNA pull-down analyses. The results were similar to those obtained for MSAs in previous studies. To further confirm that MnMYB3R1 detected by DNA affinity capture specifically binds to the *MnPPO1* promoter, rather than non-specifically, we conducted a series of validation experiments, which confirmed that MnMYB3R1 binds to the MSA in the *MnPPO1* promoter region. We demonstrated experimentally that MnMYB3R1 is an upstream regulatory factor that controls the expression of *MnPPO1*. To determine whether it positively or negatively regulates expression, we overexpressed *MnMYB3R1* in tobacco. Additionally, a phylogenetic analysis indicated that NtPPO2 is closely related to MnPPO1. Moreover, the *NtPPO2* expression level was significantly upregulated when *MnMYB3R1* was overexpressed in tobacco. These results combined with the observation that MnMYB3R1 binds to the *MnPPO1* promoter region imply that *NtPPO2* is a tobacco homolog of *MnPPO1*. The *NtPPO2* transcript level increased significantly in the *MnMYB3R1-*overexpressing plants. In the transgenic plants, the *NtPPO2* transcript levels tended to increase with increasing *MnMYB3R1* transcript levels (Figure 5b). Additionally, the PPO activity was significantly higher in the transgenic plants than in the wild-type plants (Figure 6d). These results suggested that MnMYB3R1 is a positive regulator of *MnPPO1* expression.

### 3.2. MnMYB3R1 Enhances Plant Drought Tolerance by Regulating MnPPO1 Expression

In plants, PPOs have important roles related to stress resistance [6,9,43]. However, there is relatively little information regarding the mechanism underlying the ability of PPO to increase plant stress resistance. Moreover, most of the available information indicates that the anti-bacterial or anti-insect mechanisms of PPO are related to the products of the enzyme activity [44,45]. To the best of our knowledge, there has been no published research regarding upstream molecular regulatory mechanism. Thus, we attempted to identify the upstream regulator of *PPO* expression to provide a new perspective for exploring the molecular mechanism regulating changes to *PPO* expression in response to stress. Fortunately, we first detected the MYB3R transcription factor that binds to the *MnPPO1* promoter region via a DNA pull-down assay. Most studies on the function of MYB3Rs have focused on their cell cycle regulatory role, and MYB3Rs regulate the cell cycle by controlling the expression of genes involved in the G2/M phase transition. Our results indicated that MnMYB3R1 can bind to the *MnPPO1* promoter, implying that MnMYB3R1 has other functions besides regulating the cell cycle. To explore these additional functions, we constructed a phylogenetic tree comprising MnMYB31 and other MYB3Rs with known functions. In the tree, MnMYB3R1 clustered with rice and wheat MYB3R proteins, suggesting that these proteins may be functionally similar. MYB3Rs in wheat and rice are involved in responses to drought and low-temperature stresses in addition to regulate the cell cycle [33,34,35]. Next, a qPCR analysis of *MnMYB3R1* expression revealed that *MnMYB3R1* transcripts accumulate in response to drought and ABA treatments. Regarding the effects of drought stress, the survival rates of *MnMYB3R1*-overexpressing transgenic tobacco plants were significantly higher than those of wild-type plants. Additionally, the transcript levels of most ABA pathway-related genes were significantly higher in the transgenic plants than in the wild-type plants. These results suggested that *MnMYB3R1* is likely involved in the drought stress response pathway of plants.

We also explored whether the function of *MnMYB3R1* in the drought stress response is related to *MnPPO1*. We previously determined that *MnPPO1* expression is strongly induced under drought conditions [46], and that the upstream regulatory elements in its promoter include an ABA response element. Interestingly, we observed that the PPO activity under normal conditions differed significantly between wild-type and transgenic tobacco plants, implying that *MnMYB3R1* overexpression influences PPO activity. This verified that MnMYB3R1 is a positive regulator of *PPO* expression. Under drought conditions, PPO activity significantly increased in wild-type and transgenic tobacco plants, indicating that PPO may affect drought resistance. Overall, our results indicated that MnMYB3R1 regulates *PPO* expression to help mediate drought resistance.

Considered together, our data confirmed that MnMYB3R1 is a positive regulator of *MnPPO1* expression in *M. notabilis*. Our experiments proved that MnMYB3R1 binds to the *MnPPO1* promoter via the MSA. The enhanced drought tolerance of tobacco plants harboring 35S::*MnMYB3R1* indicated that *MnMYB3R1* can regulate the expression of *PPO*, which encodes an important enzyme in the drought stress response pathway. This information will be useful for elucidating the molecular mechanism underlying the plant stress resistance mediated by PPOs.

## 4. Materials and Methods

### 4.1. Materials and Treatments

Aureobasidin A, YPDA, yeast minimal medium, SD medium, and a yeast one-hybrid kit were obtained from Clontech (Palo Alto, CA, USA). The Murashige and Skoog (MS) modified basal medium was purchased from Phyto Technology Laboratories (Shanghai, China). A dual fluorescence reporter kit was purchased from Promega (Shanghai, China). Kanamycin, acetyl syringone, naphthaleneacetic acid, 6-benzylaminoadenine, and β-mercaptoethanol were obtained from the Shanghai Biological Engineering Co. (Shanghai, China). A plant ChIP kit was purchased from Ai Meijie Technology (Shanghai, China). Strains of *Escherichia coli*, YIH Gold yeast, and *Agrobacterium tumefaciens* (LBA4404) were maintained in our laboratory. The pAbAi, pGADT7, and pLGNL vectors were available in our laboratory. The roots, stems, leaves, flowers, fruits were collected from *Morus notabilis* trees growing in Ya’an, Sichuan province, China. The collected samples were immediately frozen in liquid nitrogen and stored at −80 °C until used. Peeled *M. notabilis* seeds were germinated in culture medium with tap water until they formed roots. The resulting seedlings were transplanted into pots filled with nutrient soil, which consisted of peat moss and perlite at a ratio of 5:1 (Dahan Yuanjing Biotechnology Co., Ltd, Guangzhou, China). The nutrient soil particle sizes were as follows: 1–2 mm for fine particles, 2–3 mm for small particles, and 3–5 mm for large grains. Plantlets were grown in a PQX plant incubator at 25 °C with a 12-h light/12-h dark photoperiod until the aerial parts grew to approximately 25 cm. The seedlings were exposed to simulated drought (20% PEG 6000), salt stress (250 mM NaCl) conditions or were infected with *Botrytis cinerea* (i.e., a 5-mm agar plug containing fungal mycelium was used to inoculate mulberry leaves). Treated seedlings were incubated for 0, 8, 24, 48, or 72 h. 

Regarding the hormone treatments, leaves were sprayed with 1 mM SA, 0.1 MeJA, or 0.1 mM ABA, or were subjected to ethylene (ET) derived from 0.2 mM ethephon as previously described [47]. Leaves were harvested from seedlings at 0, 1, 2, 4, and 12 h after treatments (four pots of plants per sample in each group) and stored at −80°C for a subsequent total RNA extraction. Untreated seedlings were used as controls. All experiments were completed with three independent replicates.

To obtain anti-MnMYB3R1 antibodies, two New Zealand white rabbits were immunized with the recombinant MnMYB3R1 produced in *E. coli*. The serum of each immunized rabbit was purified on a protein A column. The antibody titer was determined by an enzyme-linked immunosorbent assay. Horseradish peroxidase-labeled goat anti-rabbit IgG (Thermo Fisher Scientific, Waltham, MA, USA) was used as the secondary antibody.

### 4.2. DNA Affinity Trapping of DNA-Binding Proteins 

The *MnPPO1* promoter fragments containing a MYB-binding site and the MSA were used to isolate the proteins binding to these *cis*-elements. Biotinylated promoter fragments were generated by a PCR with the primers listed in Appendix A. Nuclear protein extracts for DNA pull-down analyses were prepared from mulberry leaves. The biotinylated promoter fragments were immobilized on streptavidin magnetic beads according to the manufacturer’s instructions (1 μg DNA probe was immobilized on 40 μL beads with 100 μL DNA structure buffer). The mixture was incubated for 25 min at room temperature to obtain the probe–magnetic bead complex. The extracted nucleoproteins were then added to the probe–magnetic bead complex (non-biotin-labeled probe–magnetic bead complex + nucleoproteins as the control group) in a binding buffer [500 μL DNA precipitation buffer, 5 μL poly(dI.dC), 100 mM NaCl, 5 μL protease inhibitor cocktail, 9 μL EDTA, 4.5 μL EGTA, and 5 μL DTT]. The mixture was placed on a magnetic rack, and the supernatant was removed after a 30-min incubation at 4 °C. Non-specifically bound proteins were removed by washing four times with cleaning buffer (800 μL DNA precipitation buffer and 0.6 μL DTT). Finally, the target protein was eluted with 60 μL protein elution buffer containing 0.6 μL DTT, and then identified by silver staining and mass spectrometry. The reagents and buffers mentioned in this paragraph were supplied in the pull-down kit (Bersinbio Company, Guangzhou, China).

### 4.3. Cloning of the M. notabilis MYB3R1 Gene and Sequence Analyses

We retrieved *MYB3R* sequences from *Triticum aestivum* (GenBank ID: ADO32617.1), *Nicotiana tabacum* (GenBank ID: NP_001312123.1), *A. thaliana* (GenBank ID: AED90459.1 and AEE74758.1), and *Oryza sativa* (GenBank ID: XP_015616493.1). Additionally, a mulberry PPO-like gene sequence (GenBank ID: XP_010111464.2) was retrieved from the NCBI database (http://www.ncbi.nlm.nih.gov). The sequences were aligned with the ClustalX2.1 software program (https://www.rocketdownload.com/program/clustalx-431443.html). We then used the Premier 5.0 program (https://download.csdn.net/download/lhminute/7234233) to design primers to amplify and clone full-length mulberry *MYB3R1*-like genes based on the corresponding mRNA sequences. Additionally, qPCR primers for five *MYB3R1* genes were designed with the Premier 5.0 program (Appendix A). Total RNA was extracted from *M. notabilis* roots and leaves with the RNAiso Plus kit (Takara, Otsu, Japan). First-strand cDNA was synthesized with the first-strand cDNA synthesis kit (TaKaRa, Dalian, China) and used as the template for subsequent PCR amplifications. The above-mentioned primers and Ex-taq polymerase (TaKaRa, Dalian, China) were used to amplify *MYB3R1* genes from the first-strand cDNA. The PCR conditions were as follows: Initial denaturation at 94 °C for 4 min, followed by 32 cycles of 94 °C for 30 s, 61 °C for 30 s, and 72 °C for 2 min; 72 °C for 7 min. The amplified *MnMYB3R1* gene was cloned into the pMD19-T vector (TaKaRa, Dalian, China) to verify the sequence.

### 4.4. RNA Isolation and Quantitative Real-Time PCR Analyses

Total RNA was extracted from six tissues (roots, bark, buds, leaves, male flowers, and treated mulberry leaves) with RNAiso Plus (TaKaRa, Dalian, China), after which 1000 ng total RNA was used as the template to synthesize cDNA in a 25-μL reaction mixture prepared with the Prime Script RT reagent kit (TaKaRa, Dalian, China). A qPCR was completed in a final volume of 20 μL, which consisted of 10 μL SYBR Premix, 0.8 μL gene-specific primers, 0.4 μL Rox Reference Dye, 2 μL template, and 6 μL ddH_2_O. The qPCR analysis was performed with the Step-One Plus Real-Time PCR System (Applied Biosystems, Foster City, CA, USA). The mulberry *A3* gene was used as a reference control for normalizing expression data, and the relative transcript levels were calculated according to the 2^−ΔΔCt^ method [48]: ΔCt = (Ct _Target_ − Ct _A3_); ΔΔCt = (ΔCt _Treated_ − ΔCt _Untreated_); 2^−ΔΔCt^ Control value = 1; if 2^−ΔΔCt^ Target value >1, gene expression was upregulated; if 2^−ΔΔCt^ Target value < 1, gene expression was downregulated.

### 4.5. Yeast one Hybrid Assay

The *MnPPO1* promoter fragment was amplified by PCR with primers that introduced KpnI and SpeI restriction enzyme sites at the 5′ and 3′ ends. The PCR conditions were as follows: 94 °C for 4 min, 30 cycles of 94 °C for 30 s, 58 °C for 30 s, and at 72 °C for 1 min; 72 °C for 7 min. The amplified *MnPPO1* promoter sequence (bait fragment) was cloned into the pMD19-T vector (TaKaRa, Dalian, China) to verify the sequence. The bait fragment and the reporter vector pAbAi were digested with KpnI and SpeI, respectively, and then ligated at 16 °C overnight with T4 ligase. The recombinant plasmid was transformed into *E. coli* DH5α competent cells. A 100-μL aliquot of the bacterial solution was spread onto the agar-solidified LB medium (containing 50 ng/ mL Amp^+^) and incubated at 37 °C overnight. Positive clones were sequenced, after which the bait plasmid (pAbAi-*pPPO1*) was purified. Next, 2 μg pAbAi-*pPPO1* was digested with BstBI and the resulting linearized plasmid was transformed into Y1HGold yeast cells according to the manufacturer-recommended LiAC/PEG method (Clontech, Shanghai, China). 

A 100 μL aliquot of the bacterial solution was spread onto the agar-solidified SD/−Ura medium and incubated at 30 °C for three days. Five single colonies were selected. After which positive colonies were identified with the Matchmaker Insert Check PCR Mix 1. The generated recombinant yeast Y1HGold strains with pAbAi-*pPPO1* or p53-AbAi were grown on agar-solidified SD/−Ura, SD/Ura/AbA (150 ng/mL), SD/Ura/AbA (200 ng/ml), and SD/Ura/AbA (250 ng/mL) media. Plaque growth was analyzed after culturing for 2–3 days at 30 °C. The background ABA concentrations in the recombinant yeast Y1HGold (with pAbAi-pPPO1) strain and Y1HGold (p53-AbAi) strain were determined. The reporter vector pGADT7-*MYB3R1* was transferred into the recombinant yeast strains. The recombinant yeast Y1HGold (pAbAi-*pPPO1* + MYB3R1), negative control Y1HGold (pAbAi-*pPPO1*_mutation_ + MYB3R1), and positive control Y1HGold (p53-AbAi) + AD) were cultured on the SD/− Leu/AbA medium. Plaque growth was analyzed after culturing for 2–3 days at 30 °C. We mutated the MSA sequence in the *MnPPO1* promoter (AACGG to ATCGG) as a negative control.

### 4.6. Dual Luciferase Reporter Gene System

The target promoter fragment (see Section 4.2) and the pGreenII 0800-Luc vector were digested with KpnI and SpeI and then purified. Additionally, pMD19-T–*MnMYB3R1* (see Section 4.3) and pGreenII 62-SK were digested with EcoRI and XhoI and then purified. The digested fragments were ligated overnight at 16 °C with T4 ligase. The resulting recombinant plasmids were transformed into *E. coli* DH5α competent cells, after which a 100 μL aliquot of the bacterial solution was spread onto the agar-solidified LB medium (containing 50 ng/mL Amp^+^) and cultured at 37 °C overnight. Positive clones were sequenced. The bait plasmid pGreenII 0800–Luc–*pPPO1* and pGreenII 62-SK–*MnMYB3R1* were purified and inserted into *A. tumefaciens* cells, which were then used to transform tobacco leaves. After a two- to three- days of inoculation in an artificial climate chamber, tobacco leaves were collected to measure the LUC and REN signal intensities. Specifically, tobacco leaves were ground into a powder in liquid nitrogen, mixed with 200 µL of 1×PLB, and shaken for 15 min at room temperature on a horizontal shaker. The samples were centrifuged at 13,800× *g* for 15 min at 4 °C, after which 20 µL supernatant was added to a 96-well plate. Next, 100 µL LAR II was added to the wells. The plate was then shaken gently for 25 s before the LUC signal intensity was measured in a microplate reader. Finally, 100 µL of the Stop & Glo reagent was added to the well. The plate was shaken gently for 25 s before the REN signal intensity was measured in a microplate reader. The LUC/REN signal intensity ratio was calculated. The LUC/REN ratio in tobacco leaves with pGreenII 0800–Luc-*pPPO1* and the pGreenII 62-SK empty vector served as the control. The reagents and buffers used in these analyses were from the dual fluorescence reporter kit purchased from Promega (Shanghai, China).

### 4.7. Chromatin Immunoprecipitation Analysis

Mulberry leaves (1 g) were fixed in 20 mL 1% (*w/w*) methanol under vacuum conditions for 10 min. After washing twice with ice-cold 0.2 M glycine, the leaves were ground into a powder with liquid nitrogen. Chromatin complexes bound to the anti-MnMYB3R1 antibody or IgG (negative control) were obtained according to the kit instructions (https://www.epigentek.com/docs/P-2014.pdf). To quantify the precipitated chromatin, a qPCR assay was completed with gene-specific primers (Appendix A).

### 4.8. Transformation of Tobacco Cells with MnMYB3R1

The *MnMYB3R1* sequence obtained from pMD19-T–*MnMYB3R1* by digestion was directionally cloned into the SmaI and EcoRI sites of the pLGNL vector to generate the pLGNL-*MnMYB3R1* recombinant plasmid, which includes a GUS marker gene. *MnMYB3R1* expression was driven by the CaMV 35S promoter in this construct. The recombinant plasmid was inserted into *A. tumefaciens* LBA4404 cells by electroporation. The resulting *A. tumefaciens* cells were then used to transform tobacco plants according to the tobacco leaf disc transformation method [49]. Specifically, sterile tobacco leaves were collected, cut into 0.5 cm × 0.5 cm leaflets, and immersed in the *A. tumefaciens* culture (OD_600_ = 0.5–0.6) comprising cells carrying pLGNL-*MnMYB3R1*. After 10 min, the tobacco leaflets were used to inoculate the MS_1_ co-culture medium (MS + 0.1 mg/L NAA + 0.5 mg/L 6-BA), which was then incubated in the dark at 28 °C for two days. The leaflets were subsequently transferred to the MS_2_ screening medium ( MS + 0.1 mg/L NAA + 0.5 mg/L 6-BA + 100 mg/L Kan + 500 mg/L Cef ), which was then incubated at 25 °C, with a 16-h light/8-h dark photoperiod for two weeks to induce the differentiation of the resistant bud. The resistant shoots were grown to about 1 cm, after which they were cut and transferred to the MS_3_ rooting medium (half-strength MS + 0.l mg/L IAA + 100 mg/L Kan + 500 mg/L Cef). When the resistant seedlings grew to 3–5 cm, they were transplanted to pots for further growth.

### 4.9. Identification of Resistant Tobacco Plants

When the resistant tobacco plants reached the 3 to 5- leaf stage, total RNA was extracted from a few leaves. Total RNA was also extracted from wild-type leaves. The cDNA synthesized as described in Section 4.3 was used for a PCR to identified transgenic plants. The PCR conditions were as described in Section 4.3. The obtained PCR products were sequenced by Shanghai Biotech.

### 4.10. Drought Treatment of Transgenic Tobacco

Tobacco plants were grown in 20-cm pots filled with a 1:1 mixture of perlite and vermiculite under a long-day photoperiod (16-h light/8-h dark) at 22 °C. To evaluate drought tolerance, seedlings were grown in vermiculite-filled pots for two weeks with constant watering before water was withheld for two weeks. The plants were then rewatered, and the survival rates of the wild-type and *MnMYB3R1*-overexpressing plants were calculated seven days later. Plants were considered dead if all of the leaves were brown and there was no regrowth after seven days of rewatering.

### 4.11. Determination of Polyphenol Oxidase Activity

Treated or untreated tobacco leaves (1 g) were ground in liquid nitrogen. The crude enzyme extract for determining PPO activity was obtained as described in the kit instructions (http://www.solarbio.com/data/pdf/0_20180824bmwcwu.pdf).

### 4.12. Statistical Analysis

Data were analyzed with Microsoft Excel 2013, and the mean and standard error were calculated separately and plotted. The Student’s *t*-test was performed using the SPSS 13.0 program. Differences between the wild-type and transgenic tobacco plants were designated as significant (* *P* < 0.05) and extremely significant (** *P* < 0.01). 

## Figures and Tables

**Figure 1 ijms-20-02602-f001:**
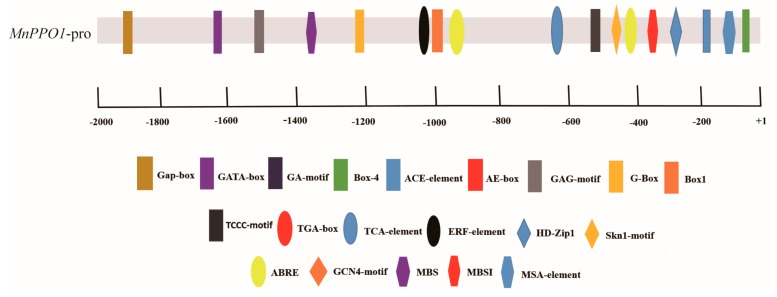
Transcriptional regulatory elements upstream of polyphenol oxidase 1 gene *(MnPPO1)*. The axis represents the +1 to −2000 bp sequence starting from the translation start site. Rectangles represent light-responsive elements, ellipses represent hormone responsive elements, hexagons represent the MYB transcription factor-binding elements and the MYB3R-binding *cis*-element (MSA), and diamonds represent other elements.

**Figure 2 ijms-20-02602-f002:**
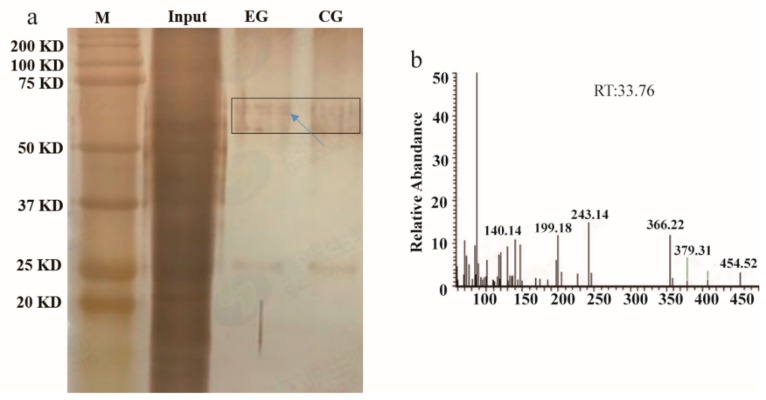
DNA affinity trapping of DNA-binding proteins. (**a**) Identification of DNA-binding proteins by silver staining; (**b**) MnMYB31 protein sequence mass spectrum. In Figure 2a, the marker represents a 250 kDa protein marker, input represents the total protein content, and EG indicates the experimental group, which comprised the proteins captured by the biotin-labeled *MnPPO1* probe. Additionally, CG indicates the control group, which comprise the proteins captured by the non-biotinylated *MnPPO1* probe. The arrow indicates differential bands. The protein bands sent to Huada Company for mass spectrometry analysis are boxed.

**Figure 3 ijms-20-02602-f003:**
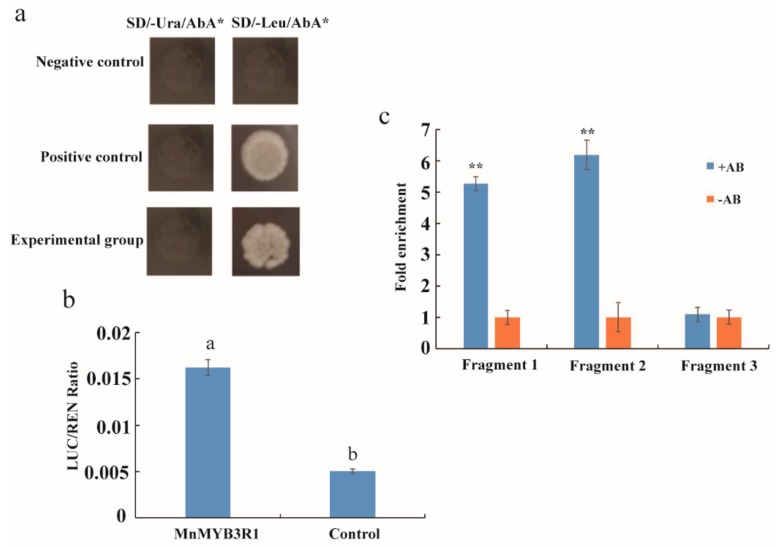
Verification of *MnMYB3R1* binding to the *MnPPO1* promoter region. (**a**) Yeast one-hybrid analysis; (**b**) LUC/REN ratio; (**c**) ChIP-qPCR analysis of the *MnPPO1* promoter. Chromatin bound to MYB3R1 was collected by immunoprecipitation with (+Ab) or without (−Ab) an anti-MnMYB3R1 antibody. Fold enrichment of the promoter region was determined by normalizing the recovery rate against that of samples immunoprecipitated without the antibody. Fragment 1 is the *MnPPO1* promoter region containing the MYB transcription factor-binding site and the MSA; fragment 2 is the *MnPPO1* promoter region containing only the MSA; fragment 3 is the *MnPPO1* promoter region without the MSA. Data are presented as the mean ± SD of three biological replicates. Significant differences from the control immunoprecipitation without the antibody were determined based on the Student’s *t*-test (** *P* < 0.01). The experimental group represents Y1HGold (pAbAi-pPPO1+ MYB3R1) recombinant yeast; the negative control represents Y1HGold (pAbAi-pPPO1_mutation_ + MYB3R1) recombinant yeast; positive control represents Y1HGold (p53-AbAi + AD) recombinant yeast.

**Figure 4 ijms-20-02602-f004:**
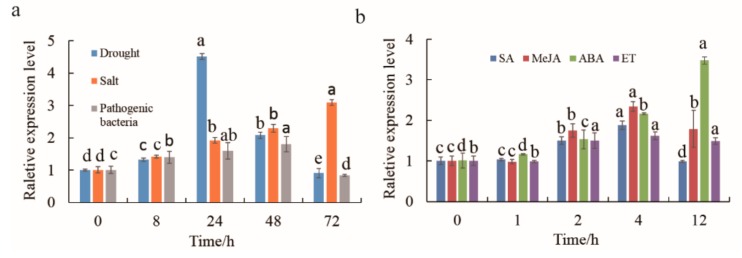
Analysis of *MnMYB3R1* transcription in *M. notabilis*. (**a**) Transcription levels of *MnMYB3R1* in mulberry leaves after drought, salt, and gray mold treatments; (**b**) transcription levels of *MnMYB3R1* in mulberry leaves after hormone treatments. The *M. notabilis* seedlings grown under the same conditions were subjected to drought, or salt stress conditions or were infected with gray mold for 0, 8, 24, 48, and 72 h. Regarding the hormone treatments, the *M. notabilis* seedlings grown under the same conditions were treated with ABA, SA, ET, or MeJA for 0, 1, 2, 4, and 12 h. Leaves collected from the treated seedlings were frozen in liquid nitrogen and stored at −80 °C for a subsequent total RNA extraction. The corresponding tissues of untreated seedlings served as controls. Relative transcript levels were determined by qPCR. Data are presented as the mean ± SD (*n* = 3).

**Figure 5 ijms-20-02602-f005:**
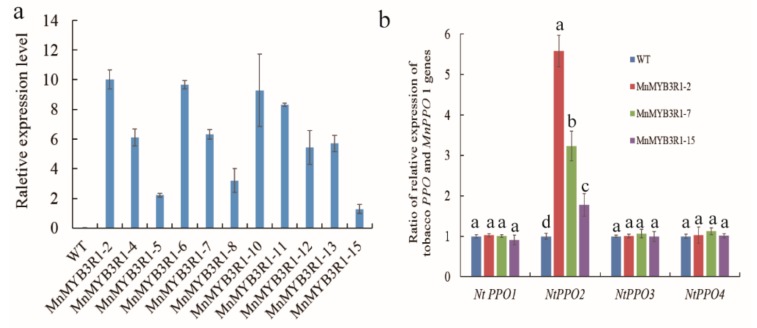
Identification of *MnPPO1* homologs in tobacco. (**a**) Quantitative results of *MnMYB3R1* expression in transgenic tobacco lines; (**b**) ratio of the relative expression of *NtPPO* in the wild-type and transgenic tobacco plants.

**Figure 6 ijms-20-02602-f006:**
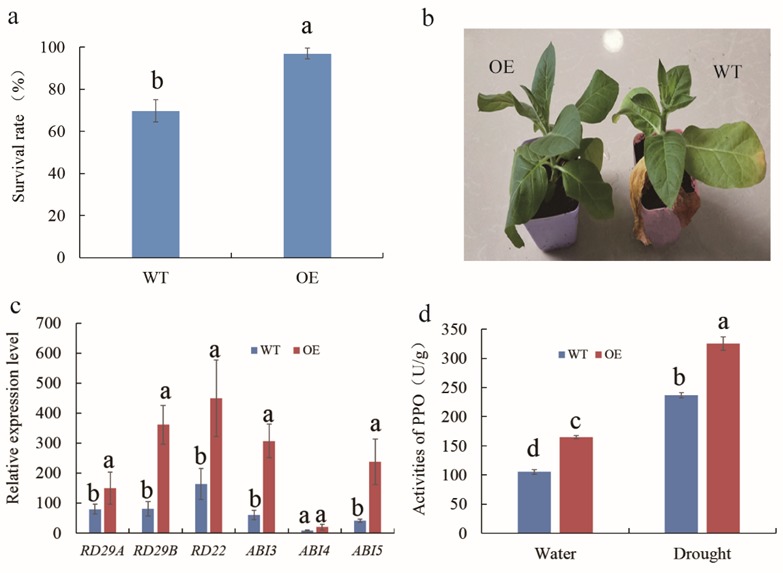
Drought tolerance experiments. (**a**) Survival rates of wild-type and *MnMYB3R1*-overexpressing transgenic tobacco plants under drought stress conditions; (**b**) phenotypes of wild-type and transgenic tobacco plants under drought stress conditions; (**c**) expression patterns of ABA signaling pathway-responsive genes in wild-type and transgenic tobacco plants under drought stress conditions; (**d**) polyphenol oxidase activities in wild-type and transgenic tobacco plants under well-watered and drought stress conditions.

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
