# Peer review of "An R1R2R3 MYB Transcription Factor, MnMYB3R1, Regulates the Polyphenol Oxidase Gene in Mulberry (*Morus notabilis*)"

_ijms, 2019, doi:10.3390/ijms20102602_

Round 1

Reviewer 1 Report

The authors already answered the comments, no any another comments. 

Author Response

Thank you very much for your time and consideration.

Minor changes that would improve the English were incorporated. 

Reviewer 2 Report

In this study, the authors have verified that MnMYB3R1 could positively regulate the expression of MnPPO1 by binding to the MSA in the promoter region, which involved in drought and abscisic acid-responsive stress response pathway. Generally, this study has the comprehensive evidences, however, I have a question about the promoter: how did the authors make sure that the promoter of MnPPO1 was about 1000bp in length? And why did the authors take the translation initiation site as a transcription initiation site? Usually, the transcriptional initiation site is marked as “-1” and the promoter region starts upstream from the transcriptional initiation site. Thus, the authors need to check and correct related data or descriptions.

Author Response

In this study, the cis-acting elements of the promoter region with a length of 2000 bp (-2000 to +1, +1 for translation start site) upstream of MnPPO1 gene was analyzed. An MSA binding site was identified at position -73. We did not amplify the full-length sequence of the 2000 bp promoter region and a promoter region of 1268 bp (-1252 to +16) was obtained with the MSA and core promoter sequences (TATA box is around -30). This 1268 bp promoter sequence was then used as a probe for the DNA pull down experiment. We have changed -1 to +1 in Figure 1, and the related explanations have been corrected at lines 13-14, 94-95.

This manuscript is a resubmission of an earlier submission. The following is a list of the peer review reports and author responses from that submission.

Round 1

Reviewer 1 Report

Manuscript IJMS472339 by Liu et al tried to prove that the MnPPO1 gene is up-regulated by MnMYB3R1 transcription factor under abiotic / biotic stresses, but it displayed low quality data and poor illustrations. It also requires additional improvement in academic writing and English editing.  It seems the authors had done comprehensive experiments. Unfortunately, most data were broken and looked like simulations. This way of doing science can be seen as inefficient at best. To prove the interaction between MnPPO1 and MnMYB3R1, fewer experiments would have been sufficient if the authors had taken a more pragmatic approach. The following are some questions and comments for consideration.

1.      The image quality of Fig. 2, 3 and S3 must be improved. Higher image resolutions and larger text font size are highly recommended. 

2.      Fig. 2a is duplicated from another publication but no citation was provided.  The primers were placed at the error positions in the diagram, suggesting that the authors might not completely understand how their experiment was designed. In Fig. 2b, the legend is incorrect. It is only a silver stained protein gel. What the EG and CG in the gel stood for was not explained. The NC indicated in the legend was not found in the gel image. No specific band can be recognized. What is the marked rectangle box? In Fig. 3c, the resolution is too low to provide any meaningful information for readers.

3.      For Fig. 3b and the dual reporters experiment, data from multiple time points should be collected. The method on how REN and Luc were assayed was not provided in Methods section. In Fig. 3c, the ChIP experiment should be done with the transient expression system to investigate the interaction between MSA element and MnMYB3R1. Information on how the authors made the MnPPO1 mutation in Fig. 3d must be provided in Methods. 

4.      Methods on drought and salt treatment of mulberry plants must be described in details. Transcription level of MnPPO1 in the mulberry plants under the same abiotic and biotic stresses should be assayed to prove it is an inducible gene under the same stress conditions.

5.      Detailed information on the transgenic plants must be provided. The authors should answer: How many transgenic lines were generated? How were they selected, kept and amplified?  How were the copy numbers and expression levels of the transgene determined? What were the phenotypes of the transgenic plants?  Which transgenic lines were used for further analysis of the NtPPOs?

6.      In Fig.5, there appeared to be a dosage effect on the NtPPO2 expressions by MnMYB3R1. The authors should analyze their data carefully for a clarification.

7.      Line 178, which ABA pathway-related genes were assayed should be summarized in the results.

8.      Line 190, the authors should find out how the increase of the PPO activity was correlated to NtPPO1 to 4. The transcription level of the NtPPO1 to 4 genes under the drought condition must be assayed.

9.      The authors suggested that MnMYB3R1, OsMYB3R2 and TaMYB3R1 may share similar functions. Please discuss these function as well as the associated literatures in Discussion section.

10.  Please discuss the functions of NtPPO genes and their inducibility in Discussion.  

11.  Line 232, please include the citations of the previous studies.

Reviewer 2 Report

The manuscript entitled " An R1R2R3 MYB transcription factor, MnMYB3R1,regulates the polyphenol oxidase gene in mulberry (Morus notabilis)” by Dan Liu, Shuai Meng, Zhonghuai Xiang, Guangwei Yang, Ningjia He, is a good attempt towards the mulberry polyphenol oxidase 1 (MnPPO1) regulation by MnMYB3R1 transcription factors. The authors verified the MnMYB3R1 binding to the promoter region of MnPPO1 gene trough the MSA element. Moreover the authors assessed that MnMYB3R1 is an upstream, positive regulatory factor of MnPPO1 gene. Finally the authors proved that MNMYB3R1 is likely to be involved in the drought stress response of in genetically modified tobacco plants. The work is of interest for the Journal. Only the following minor revisions are required:

Please, change in italics all “Morus notabilis” in the manuscript.

Please, let change in italics all the gene names in the manuscript.

Results

Could, please, the authors report the statistical tests applied to assess the significant differences in MnMYB3R1 genes expression in mulberry leaves and transgenic lines of tobacco, in MnPPO genes expression in tobacco and in PPO activities in wt and transgenic MnMYB3R1 tobacco under well-watered and drought stress condition?

Materials and Methods

Line 303: Please, add the ClustalX2.1 software citation

Line 304: Please, again add the Premierv.50 software citation

Line 310: Please, add the Ex-Taq polymerase supplier info

Line 323: Please, add the 2-D∆Ct method citation

Line 379: Please replace “Tobacco plants were was grown…” with “Tobacco plants were grown”

Reviewer 3 Report

This manuscript entitled “An R1R2R3 MYB transcription factor, MnMYB3R1, regulates the polyphenol oxidase gene in mulberry (Morus notabilis)”reports about novel transcriptional regulation mechanisms in Mulberry Polyphenol oxidase 1 gene under regulated by MYB transcription factor.
The polyphenol oxidase is an important enzyme for modification of phenolics in plants although their function under the stress conditions is less unknown.
I think results documented in this manuscript are valuable for clarifying these questions and increase in knowledge of this field.

I would like to point out some issue in this manuscript.

Major point
-I would like to suggest that the authors need to discuss about more details.
The authors wrote the conclusion of their results in discussion part, but it is not enough to show the significance of this research. They should discuss the novelty of this research by citing more previous reports in discussion part.

Minor points
There are some mistakes in this manuscript.
- All gene symbols and some chemical terms should show the Italic characters.
-In legend of Figure 2 (b), please indicate the meaning of EG and CG.
- In Figure 6b, please indicate which plant is OE or WT in picture.

Reviewer 4 Report

This manuscript revealed transcription factor MnMYB3R1binds to the promoter region of MnPPO1 under stress response. The authors have demonstrated that MSA cis-element play an important role in the binding. But I have some question or advise on your advise.

Line 78: This transcription factor is a 3RMYB that is known to play roles in regulation of the cell cycle.   Reference? Or I don't see any evidence in your manuscript.

Line 110-112: Fig. 3d should follow Fig. 3a and b

Line 116-117: Did fragment one, fragment two, fragment three have overlap?

Line 132-147: Did you see the expression of MnPPO1 in the different stress treatments .

Line 163-164: Fig. 5a?  I think Fig. 5b

Line 170-179: Did you see the expression of MnPPO1 in the ABA treatment .

Line 186-191: I think Under normal and drought conditions, both PPO activity was significantly higher in transgenic tobacco than in wild type (Fig. 6d). Can you explain why?

  Also, English writing needs further editing.